# Development of a Custom-Made 3D Printing Protocol with Commercial Resins for Manufacturing Microfluidic Devices

**DOI:** 10.3390/polym14142955

**Published:** 2022-07-21

**Authors:** Francesc Subirada, Roberto Paoli, Jessica Sierra-Agudelo, Anna Lagunas, Romen Rodriguez-Trujillo, Josep Samitier

**Affiliations:** 1Nanobioengineering Group, Institute for Bioengineering of Catalonia (IBEC), Barcelona Institute of Science and Technology (BIST), 12. Baldiri Reixac 15-21, 08028 Barcelona, Spain; fsubirada@ibecbarcelona.eu (F.S.); rpaoli.138zb@simplelogin.co (R.P.); jsierra@ibecbarcelona.eu (J.S.-A.); alagunas@ibecbarcelona.eu (A.L.); romen.rodriguez@ub.edu (R.R.-T.); 2Biomedical Research Networking Center in Bioengineering, Biomaterials, and Nanomedicine (CIBER-BBN), Av. Monforte de Lemos, 3-5, Pabellón 11, Planta 0, 28029 Madrid, Spain; 3Department of Electronics and Biomedical Engineering, University of Barcelona, Martí i Franquès 1, 08028 Barcelona, Spain

**Keywords:** microfluidics, 3D printing, stereolithography, additive manufacturing, photo-curable polymers

## Abstract

The combination of microfluidics and photo-polymerization techniques such as stereolithography (SLA) has emerged as a new field which has a lot of potential to influence in such important areas as biological analysis, and chemical detection among others. However, the integration between them is still at an early stage of development. In this article, after analyzing the resolution of a custom SLA 3D printer with commercial resins, microfluidic devices were manufactured using three different approaches. First, printing a mold with the objective of creating a Polydimethylsiloxane (PDMS) replica with the microfluidic channels; secondly, open channels have been printed and then assembled with a flat cover of the same resin material. Finally, a closed microfluidic device has also been produced in a single process of printing. Important results for 3D printing with commercial resins have been achieved by only printing one layer on top of the channel. All microfluidic devices have been tested successfully for pressure-driven fluid flow.

## 1. Introduction

Microfluidics is an important tool for biomedical engineering, offering advantages in reagent consumption, waste generation, and process integration. Many different methods are used to produce microfluidic devices such as injection molding, micromachining, micro milling, hot embossing, and soft lithography. However, these processes can be time consuming, expensive, and challenging for entirely three-dimensional designs. In addition, they generally must have a dust-free environment (cleanroom) to ensure the production of devices without errors [1,2].

In recent years, additive manufacturing, generally known as 3D printing, which can create solid three-dimensional (3D) objects layer-by-layer under precise digital control, has emerged as a promising alternative to develop microfluidic devices [3]. From the several 3D printing approaches used in microfluidic fabrication, stereolithography (SLA) based on the use of transparent photocurable resins to build well-defined microchannels, appears as one of the most suitable 3D technologies when compared to the alternatives. For instance, in Fused Deposition Modeling (FDM), also known as thermoplastic extrusion, the sizes of the extruded filaments are larger than typical channels used in microfluidics making it difficult to perform channel intersections due to the joining filament ends that can cause leaks [4,5].

SLA was the first form of 3D printing invented in the 1980s. Though nowadays Hull is considered the first inventor of SLA [6], a patent was previously filed in 1984 by three French inventors [7]. SLA allows for the assembly-free simultaneous production of quasi-arbitrary 3D shapes in a single polymeric material from a liquid photo-resin precursor by means of a digital light processing (DLP) projector. The DLP projector forms the projected image using a Digital Micromirror Device (DMD), a rectangular array of micromirrors which can be rotated individually. In DLP-based SLA, the XY-resolution is limited by the size of the projected pixels, and the Z-resolution is determined by the resolution of the Z-motor. The effective polymerization process and the light scattering affect the resolution achieved in both width and height [8,9].

In the DLP SLA printing process, the printed object is generated upside-down, hanging from a build platform that can be moved across the Z (vertical) axis. The process starts by lowering the platform and submerging it in a resin-filled vat with a transparent bottom, usually made of glass. Through precise motion, only a thin, few-microns-thick layer of photocurable resin is left between the platform and the vat bottom. After exposing the UV light, the platform is then moved up and down again, to the level of the next layer [10].

In general, exposure dose (D) can be regulated through two parameters: projector irradiance ($E_{e}$) and exposure time (Δt) [11,12]:(1)$$D=E_{e}\cdot \Delta t$$

To understand SLA resolution, it is important to differentiate between dimensional accuracy when printing features on top of the object versus printing on cantilever-like structures (i.e., closed channels). The difficulty of printing closed microfluidic channels is shown in Figure 1, where the different steps for printing a cantilever structure are shown.

For printed features on the top of the object, ideal XY resolution is mainly limited by the projected pixel size, the absorption spectra of the photo-resin, and the diffusivity of the reactants, while ideal Z resolution is limited by the minimum step that can be achieved by the building platform [13]. In previous studies, it has been demonstrated that SLA has a better resolution in the XY-plane than in the Z-direction [14]. In comparison, printing cantilever structures become more challenging. When printing the roof of a cantilever structure, light tends to penetrate across the resin into the previous layers, affecting the Z-resolution (Figure 1). Furthermore, the minimum cross-sectional area of a microchannel that is attainable by SLA depends not only on the pixel resolution, but also on the type and viscosity of the resin, which has to be effectively drained from the channels post printing [15]. A transparent resin is required for microfluidics in order to view the fluid within channels under a microscope. To polymerize transparent resins, a UV light source is required. This work aims to analyze and compare the resolution and capacity to generate functional microfluidic devices with transparent commercial resins using UV light. Working with a DLP projector with a wavelength peak at 384 nm, we have chosen two resins with similar absorption coefficient at a wavelength of 384 nm and we compared them with a third resin with an absorption coefficient 2.5 times higher. With this technique, which has the capability to make channels, bifurcations, cavities, or other useful features, microfluidic devices can be obtained [16,17].

This paper aims to give a brief overview in the field of 3D printing with commercial resins, an analysis of their absorbance, resolution in width and height, and rugosity for all the three chosen resins. After the characterization of the resins, a functional microfluidic device consisting of a spiral that mixes the fluid and separates particles by their diameter was made as proof of concept using different 3D printing approaches.

## 2. Materials and Methods

### 2.1. 3D Printer

The following subsection will focus on the election of the pieces chosen to customize the 3D printer: these include the DLP UV light projector, the Z motors, the plate holder, the resin container, and the open-source solution found for the hardware.

On the hardware side, the open system consists of two parts: a Raspberry Pi^®^ embedded computer and the open-source Reprap Universal Mega Board with Allegro driver (RUMBA) controller board. RUMBA board was released under GNU Public License (GPL) v2.0 by reprapdiscount.com. Raspberry Pi^®^ runs NanoDLP front-end and control software, downloaded from nanodlp.com. The software enables customization of the whole printing process while providing a slicer and a remotely accessible web interface. The software communicates with the open-source Marlin firmware which runs on the RUMBA board and controls axis movements. Marlin firmware is released under GPL v3.0. Marlin firmware was configured to control the two Z-axes through two different motors and two optical end-stops based on tcst2103 sensor for reliable alignment at each power-cycle.

The projector is Wintech PRO6500 based on DLP6500 Digital Micromirror Device (DMD) by Texas Instruments^®^ (Dallas, TX, USA). It is a UV DLP LED system with 1920 × 1080 resolution and a projected pixel size of 50 μm at fixed working distance. Peak wavelength is 384 nm. The projector is equipped with standard High-Definition Multimedia Interface (HDMI) and DisplayPort connectors for video input and Universal Serial Bus (USB) and Inter-Integrated (I2C) ports for projector control. A custom program was written to control projector LED status and intensity over USB protocol. The program was configured in the frontend (NanoDLP).

The resin container is “Resin Vat Anodized Aluminum with a Fluorinated Ethylene Propylene Film (FEP) from ANYCUBIC”.

The plate holder is a custom-designed piece to hold a standard 50 × 75 mm glass slide with a vacuum system by alignment pins into the platform. This one is matte black to avoid light reflection.

### 2.2. Resins Used and Absorbance Test

Three commercial resins were used: FREEPRINT^®^ ortho 385 nm from DETAX [18], KeyOrtho IBT from Keyprint [19], and PlasCLEAR from Asiga [20] with different mechanical properties according to the datasheets. Two of the resins, from Detax and Asiga, are hard resins after curing with a Young Modulus (E) of 1650 MPa and 1915 Mpa, respectively. The Keyprint resin is flexible after curing with an E value around 15.5–31 MPa. From the fabricants’ information, DETAX and Asiga use the same photo-initiator Diphenyl(2,4,6-trimethylbenzoyl)phosphine oxide (TPO) but Asiga shows 2.5 times higher absorbance value at 384 nm due to the difference in the absorber concentration. Both will be compared with the Keyprint elastomeric flexible resin which has a similar absorbance coefficient at 384 nm than Detax (see Figure 2). These three resins, which are often used in the dental industry, were selected for being biocompatible, transparent, and compatible with our DLP projector’s wavelength (384 nm). The absorbance test was conducted with the Infinite M200 PRO Multimode Microplate Reader from Tecan. The experiment was designed introducing 10 µL of every resin in three holes in a 96-well plate, with a pipette for viscous liquids. Then the absorbance was calculated three times and an average was done, subtracting the result obtained for a blank well. Considering the dimensions of the plate reader well, a volume of 10 µL was filling the resin to a height of 320 µm. Given that the nominal height of the 3D printed layers is 100 µm, the absorbance results were divided by a factor of 3.2.

### 2.3. 3D Printing Protocol

All the 3D prints were done in a yellow-light room, to ensure the resin does not get any UV light exposure other than the DLP UV projector. The printing chamber was preheated to 30 °C with an air heater to have constant properties of the resin.

Once the Raspberry Pi, the Z motors, and the DLP projector were turned on, we made sure that the vat contained enough resin for the piece that was going to be printed and that the 50 × 75 mm glass slide, where the 3D printed piece will be attached, was correctly aligned. Afterwards, from outside the room (important for avoiding UV radiation), the 3D printing process was executed from the software NanoDLP. This software automatically slices the drawing previously uploaded and executes the printing process based on the layer thickness, time exposure, and light intensity between other settings which must be programmed on the software.

Once the printing process was over, the piece was washed three minutes in 2-Propanol and post-cured during five minutes with UV light in the Washing and Curing Machine from CREALITY.

The optimal printing parameters were obtained using dose calibration experiments. Motives were exposed to UV light for different time steps ranging from 1 s up to 5 s for Detax and Keyprint and from 1 s to 30 s with Asiga as a longer exposure time was needed.

### 2.4. Chip Manufacturing

Three different techniques were used to produce a microfluidic device involving 3D printing. The first one and the most common was 3D printing a mold and creating the channels with a PDMS replica. This approach started with a silanization process of the mold, consisting in one minute of plasma cleaner at 30 W with constant pressure of 0.8 Torr. With the activated surface, a drop of Trichloro(1H,1H,2H,2H-perfluorooctyl) silane (CF_3_(CF_2_)_5_CH_2_CH_2_SiCl_3_) was deposited on top of a glass slide or a Petri dish inside a vacuum desiccator for one hour. Afterwards, to create the PDMS replicas, the 3D printed mold was covered with PDMS with a ratio of 10:1 of curing agent and left on the oven at 85 °C overnight. After unmolding the PDMS, the next step was to wash the PDMS replicas with soap and water, and ethanol, as well as the glass slides which were cleaned with ethanol, acetone, and 2-Propanol. Finally, the cleaned PDMS and glass were activated 30 s with the plasma cleaner at 30 W and pressure of 0.8 Torr, and the two parts were bonded. The final device was tested with water with colorant to check the absence of leakage between the microchannels. The result was a microfluidic device with the architecture of the 3D printed configuration.

The second way to create a microfluidic device was printing a part with an embossed microfluidic channel. To create a closed channel, the best option was to also 3D print the cover and assemble both pieces by after-curing them together.

The third and most innovative way to do it was printing the device in one single process. For this, it is necessary that no obstructions of the channel occur while printing the cover due to the residual UV light, and to flush the uncured resin out of the channel before post-curing it. Therefore, it is relevant to study how many layers could be printed on top of the channel, as a cover, before obstruction takes place.

### 2.5. Dimensional Characterization

To determine the dimensions of the resulting devices, we used an optical microscope (Nikon Eclipse L150, Nikon, Tokyo, Japan). To determine the height of the features, we used a profilometer (DEKTAK XT, Bruker, Billerica, MA, USA). To determine the section of the 3D printed channels, we used an inverted microscope (Eclipse Ti, Nikon, Tokyo, Japan).

### 2.6. Projector Irradiance Calibration

Projector irradiance was measured using a Power and Energy meter USB interface with S302C Thermal Power Sensor (Bandwidth: 190~2500 nm, Maximum 2 W, Sensor diameter 12 mm) from Thorlabs (Newton, NJ, USA). The sensor was fixed to the build platform adapter of the printer and brought to focus distance.

Measurements were performed at different projector PWM setting value (PWM = 20, 45, 50, 70, 100, 150). Irradiance was calculated for 2 different configurations: (i) no vat installed, (ii) vat with new FEP, shown in Figure 2, where the irradiance of the DLP projector is shown.

As shown in Figure 2, the FEP absorbs less irradiance for the higher values of LED intensity, but the irradiance is nearly the same at 35 PWM, the value selected for our 3D prints.

### 2.7. Rugosity Test Atomic Force Microscopy

Exterior surface roughness was calculated from atomic force microscopy (AFM) measurements performed on a Dimension 3100 AFM instrument (Veeco Instruments, Plainview, NY, USA) equipped with a rectangular silicon AFM tip (Budget Sensors, spring constant k = 40 N/m and a resonant frequency ν = 300 kHz) and operated in tapping mode at room temperature in air. Topographic images obtained from the scanning of 50 × 50 μm rastered areas were used for root-mean-square calculations performed with WSxM software (Nanotec Electrónica, Tres Cantos, Spain) [21]. At least three positions were analyzed per sample.

In most additive manufacturing processes, the addition of discretized layers creates a sidewall roughness (Figure 3), which is inaccessible by AFM. To study the surface roughness of the sidewalls, we used a prediction model. According to ASME B46.1 [22], the surface quality can be described by the average surface roughness, which is defined as the summation of the peaks and valley areas divided by the evaluation length. This quality parameter can be calculated by the following Equation (2), and also shown in Figure 3, where the variables W and A for calculating the sidewall roughness are illustrated:(2)$$R_{\alpha }=\frac{A}{W}$$

## 3. Results

We started with an absorbance test, to confirm that the resin matched with our 3D printer projector wavelength. The following task was defining the resolution of the 3D printer, using a test sample designed to find the width and height resolution. Upon completing the prior tasks, we used the 3D printer to produce microfluidic devices with three different approaches. Rugosity tests were performed to know the microfluidic characteristics of the obtained devices.

### 3.1. Absorbance Characterization

In order to ensure the three tested resins from Detax, Asiga, and Keyprint were optimal for our DLP projector, with a peak at 384 nm, an absorbance test was made. The result of this test is shown in Figure 4, where the absorbance of the three resins is represented between 350 nm and 440 nm.

As appreciated in Figure 4, all three resins are appropriate to be polymerized with our UV light projector, which projects light at 384 nm. For the Detax resin, a peak in 384 nm is shown in Figure 4, which is optimal for our 3D printer. The Keyprint resin does not have a peak at 384 nm but has a very similar absorbance level than the Detax resin so it was also tested. On the other part, it can be observed than the Asiga resin has about 2.5 times more absorbance than the Detax one, so it was also tested.

### 3.2. Width Dimensioning

To determine the XY resolution of the technique, we used a parametric design as a model, and we printed it with the three different resins (Detax freeprint ortho, Asiga plasclear, and Keyprint keyortho IBT) at three different heights (200 µm, 400 µm, and 600 µm). Figure 5 shows a representative example of the result obtained with Asiga resin at 600 µm.

The three variable parameters introduced by the model are the pillar height, pillar width, and separation between pillars. These parameters were chosen because they were the parameters considered relevant to observe which types of features could be achieved with each resin and the extent of printing resolution. In each print, the height of all the pillars remains constant, and the width and separation changes depending on the row and the column, respectively. Sixteen subgroups of pillars are placed in the design, grouped in a 4 × 4 array. Starting from left to right in Figure 5a, in column 1, the width is half of the height; in column 2, the width is the same as the height; in column 3, the width is two times height, while in column 4, the width is three times the height. From top to bottom, in row 1, the separation between pillars is half of the width of the pillar; in row 2, the separation is the same as the width; in row 3, the separation is the double as the width, while in row 4, the separation is three times of the width. The three designed heights are 600 µm, 400 µm, and 200 µm, which means the biggest pillar is designed with a width of 1800 µm and the smallest one 100 µm. All the pillars are designed with a length of 6 mm.

The aim of the study was to compare the best resolution obtained for each of the resins while adjusting the dependent printing parameters. Layer thickness, light power, and time exposure were adjusted until the optimal settings were reached, as shown in Table 1. These parameters were obtained empirically after doing the experiments with a variety of settings until reaching the conditions that minimize the fabrication errors. However post-curing time or hatching direction were not considered dependent printing parameters. The adjusted parameters are the exposure time for the burn in layers and the normal layers. It must be longer for the burn in layers because these layers are the first ones of the print, and they need to attach to the glass slide and support all the printing process. The layer thickness, which is the distance the Z motors move up each layer, must be defined. Finally, the light power of the UV projector must also be defined.

As the exposure time needed to polymerize each resin is different, it means the different resins have different critical energy (Ec) values. Ec is the value for the minimum amount of energy required to form a solid layer and therefore, the minimum amount of energy required for 3D printing. This value varies according to the type of photo-initiator and its concentration.

Combining the results of the time needed to polymerize each resin presented in Table 1 and the absorbance results shown in Figure 2, it was seen the Asiga resin needs almost 17 times more energy to polymerize than the Detax resin, because it needs 6.7 times more of light exposition with an absorption 2.5 times higher. Instead, Keyprint resin needs only 1.6 times more energy to polymerize, as Detax resin has an absorbance 1.25 times higher and Keyprint resin needs double of the time to polymerize.

Using the parameters shown in Table 1, the smallest features achieved with each resin are reported in Figure 6, where the smallest structure obtained with a good definition and a good separation between features is shown. Additionally, the structure where the features start to have an imperfect separation is shown.

The width error was studied for all the three resins for all the different structure configurations and a clear trend was identified. The presented error was obtained measuring the size of the objects in the printed result and comparing the result with the designed size. As seen in the following graphs shown in Figure 7, the resolution error increases as the size of the structure and the separation between them decrease.

In the graphics shown in Figure 7, the X-axis corresponds to the width of the feature, going from half the height in the left to three times the height in the right. The Y-axis instead corresponds to the separation between the features, going from half of the width in the front to three times the width in the back. The Z-axis represents the error calculated in the height of the features in percentage.

The results obtained for the three resins with the designs of 400 µm height is shown in Figure 7, because it is the smallest height achieved with the Keyprint resin and was considered relevant to show the graphic for the same height in all three resins.

For the Detax resin, the error goes from 0.65% in the worst case to 0.02% in the optimum scenario. For the Asiga resin, the error goes from 15.08% to 0.69%. For the Keyprint resin, the error goes from 16.45% to 0.23%.

### 3.3. Z Dimensioning

The height definition was studied using the same 3D printed parts shown in Figure 5. The height of every feature of the 3D printed part was measured with the profilometer and the average was calculated. The height error was obtained by the comparison of the average obtained with the designed height. The result is shown in Table 2.

As seen in Table 2, the percentual error increases if the designed height decreases. Additionally, these results show that Detax is the resin with less height error, followed by Asiga, and then the one with more error is Keyprint. This could explain why with the Keyprint resin, the design with 200 µm height could not be obtained.

### 3.4. Microfluidic Devices Fabrication Using 3D Printed Molds or Assembling 3D Printed Parts

Microfluidic applications as mixers and particle separators have emerged as powerful tools in cell sorting and classification for biological applications. We designed and developed 3D printed molds for creating channels with the shape of a spiral, shown in Figure 8. The spiral channel with 1 mm of width and height of 600 µm was previously simulated in COMSOL Multiphysics^®^ for separating bigger particles of around 70 µm and smaller particles of 20 µm due to the effect of inertial forces in microfluidic channels [23]. Figure 8a shows the embossed extrusion to create an open channel, which is used for creating PDMS molds and can be observed in Figure 8b. Figure 8b shows the PDMS replica bonded to glass and tested with dyes for further confirmation of the absence of leakage between the channels.

As shown in Figure 8b, in addition to its working principle for separating particles by their diameter, the spiral design also works as a microfluidic mixer, i.e., the blue and red fluids are completely mixed at the device outlets. This mixing effect is obtained thanks to the appearance of a Dean flow due to the channel curvature at the operation flow rates (6300 µL/min for the blue fluid flow and 700 µL/min for the red fluid flow) [24].

As a second approach, we used the open channel shown in Figure 9a and attach a 3D printed cover with connectors, as shown in Figure 9b, to have a closed microfluidic channel. Moreover, the devices were tested using syringe pumps and two different dyes were passed through the channel confirming the absence of leakage as seen in Figure 9c.

In Table 3, height and width errors are presented for the designs shown in Figure 8a and Figure 9a, for each type of resin.

We can observe that the error is slightly higher in the embossed designs compared to the extruded ones and reaffirm that the height error is higher than the width error. Very similar errors were obtained for Detax and Asiga resins, while Keyprint resin had a slightly higher error. Thus, if the flexibility is not relevant, the Detax and Asiga resins are a better option for manufacturing microfluidic devices when higher precision is desired. However, if a flexible behavior is needed for the device, the Keyprint resin is optimal, even having a higher resolution error.

### 3.5. Microfluidic Devices Fabrication in One-Step

The third option was printing all the microfluidic chips in one single 3D printing process. For 3D printing a microfluidic device, it is essential to obtain channels that remain unobstructed while printing the top cover, because the UV light can cross the exposed layer and go through the piece. The other important condition is being able to control the channel section dimensions. For these reasons, before trying to print the spiral device in one single 3D printing process, a design with channels of 1 mm width and 500 µm height with a different number of layers in the cover of the channel (1, 2, 4, 8 layers of 100 µm thickness each) was printed to see if closed channels could be obtained, and how many layers could have their cover before the uncured resin remaining in the channel became polymerized. Figure 10a,c,e show the 3D printed part with Detax, Asiga, and Keyprint, respectively, and Figure 10b,d,f show a zoomed view of the channel with a cover of 100 µm, also for Detax, Asiga, and Keyprint resins, respectively.

Firstly, in the result with the Detax resin shown in Figure 10a, only the channels with one and two layers as a cover remained unobstructed, while the channels with four and eight layers as a cover ended up being obstructed. Secondly, in the result with the Asiga resin shown in Figure 10c, all channels remained unobstructed. So, channels with a cover made from up to eight layers of 100 µm can be 3D printed. Finally, in the result with the Keyprint resin shown in Figure 10g, none of the channels had a favorable result. The cover of the first channel shown in Figure 10f was too delicate and broke during the cleaning process, and all the other channels with more than one layer in the cover were obstructed with polymerized resin.

The thickness of the cover of all the channels was measured for the three resins as shown in Figure 11, together with the designed dimensions.

From Figure 11, we can observe that the final thickness of the top layer of the channel is related with the absorbance of the resin. A table with the values of the thickness for every number of layers with the three resins is shown in Appendix A.

Horizontal as-printed surface roughness was calculated from atomic force microscopy (AFM) with topographic images obtained from the scanning of 50 × 50 μm areas as explained in Section 2.7. The test has been performed for the design of the spiral printed channel shown previously in Figure 8. After analyzing a 50 × 50 μm area for the Asiga design, the roughness result was 40.61 nm. Instead, the Detax result for the same test was 66.37 nm, while the Keyprint one was 179.45 nm. As an example, the image obtained with the AFM for the Asiga resin is shown in Appendix B.

As shown in Figure 10, the only resin with a pronounced sidewall roughness is Asiga. Following Equation (2), the Asiga sidewall roughness is 27.64 µm. On the other hand, for the Detax and Keyprint resins, we appreciate very smooth sidewalls, because of their lower absorbance. We calculated a sidewall roughness under 3 µm for both Detax and Keyprint resins.

The last roughness to analyze is the internal superior part of the cover channel. In this case, Asiga has the smoother roughness while having the higher absorbance. For the Asiga superior part of the channel, the roughness is 5 µm. For the lower absorbance resins the roughness is higher, with the Detax resin the roughness is 20 µm, and for the Keyprint resin it is 25 µm.

Based on the results shown in Figure 10, we proceeded to 3D print the spiral from the previous Figure 8 and Figure 9 with closed channels, only with the Detax resin. This decision was made by discarding the other two resins for different reasons. The Asiga resin was discarded because of its vertical wall rugosity inside the channel, shown in Figure 10d, which would disturb the particle flow movement inside the channel. The Keyprint resin was discarded because for one layer of 100 µm, the channel cover will not resist the cleaning process, and with a cover of more than one layer, the channel will be obstructed when the projected UV light reaches the uncured resin of the channels leading to the photo-polymerization of the remaining resin. The result is shown in Figure 12, where the 3D printed spiral is shown.

For using the aforementioned method, which is very innovative and has a lot of potential, only one layer was being printed on top of the channel. The major drawback when printing only one layer on top of the channel is that connectors cannot be printed on top of the inputs and outputs, otherwise the channel would get polymerized and obstructed under the connector. External connectors must be attached in the inputs and outputs before any microfluidic application can be done.

## 4. Discussion

When the build platform is lifted between layers, the cured layer is separated from the base media through a dynamic process. This process can produce delamination errors, when a given layer is incompletely cured and might remain stuck on the bottom of the resin tank. To avoid this problem, it is necessary to ensure that the exposure will completely cure the current layer to increase the bonding force with the previous layers. However, this process causes a reduction in dimensions when producing cantilever or closed channels for microfluidic devices. In this work, we optimized the printing parameters of three commercial resins, and we confirm in agreement with previous studies that stereolithography has a better resolution on the XY-plane (width) compared to the Z-plane (height). For the width resolution, it has been demonstrated that the percentage error increases while the feature size and the separation between features decrease. Thus, the bigger the element is and the more separation it has with other aspects, the better the width resolution will be.

Considering that it has been demonstrated that the building orientation of the 3D printed part can affect their mechanical properties in many 3D printing technologies [25], in SLA applied to microfluidic devices, this effect is less relevant given that the microfluidic devices dimensions in X and Y are always higher than that in Z-direction. So, printing the layers in the XY-plane reduce the build time as the total number of necessary layers is significantly reduced. In addition, the anisotropic mechanical properties between XY-direction and Z-direction is less relevant considering that in water based fluidic applications the liquid pressure is isotropic.

As regards to the microfluidic devices fabrication, the size of the channel and the roughness of the channel are key parameters to consider. The sidewall and internal layer roughness is controlled by a staircase phenomenon and the critical polymerization exposure time characteristic of the resin. The micromirrors pattern of the projector act as micro-optical apertures, which diffract the reflected light. This process produces an irradiance distribution for each micromirror that can be fitted as a Gaussian distribution. The contribution of all micromirrors in the two-dimensional reflected surface introduce a staircase phenomenon in the vertical surface of the microfluidic channel. These phenomena decrease if the thickness of the layers deposited decreases or if the absorbance of the resin is lowest for a same thickness deposited value. For this reason, the channels obtained with Detax resin show a better sidewall definition. However, the decrease in the absorbance of the resin reduces the accuracy in the thickness of the layer deposited, increase the roughness of the internal surface of the cover layer, and reduce the number of layers that could be printed on top of the channel due to an overcuring effect.

As shown in Table 2 and Table 3, the results obtained with the Keyprint resin have a higher error in both width and height than the results obtained with the Detax and Asiga resins. For this reason, unless the flexible behavior of the Keyprint resin is desired, the Detax and Asiga resins are more suited options for creating microfluidic channels and molds for PDMS replicas, due to their higher resolution.

The most attractive advantage of printing closed channels, as shown in Figure 12, is that no human task was needed after finishing with the 3D printing and cleaning process, so it avoids any type of human error while assembling the device. However, the drawback of this method is that the superior layer of the channel has a higher roughness compared with the methods involving PDMS replicas (Figure 8) or assembling different 3D printed parts (Figure 9). Geometrically, the channels obtained with PDMS replicas or assembling 3D printed parts are identical, as PDMS creates very precise replicas from the 3D printed molds. Then, the major difference is the higher deformability of the PDMS replica (Young modulus around 1–2 MPa) versus the harder 3D printed pieces.

For creating devices by assembling different 3D printed parts, Keyprint resin is the best resin to print covers, because it remains sticky even after cleaning with 2-Propanol and remains assembled to another 3D printed piece if post-cured together.

The results shown in Figure 4 regarding the absorbance of the resins and the results obtained while printing closed channels shown in Figure 10 are in good agreement. As shown in Figure 4, the Asiga resin has the highest absorbance, and consequently, it is the resin that allows printing more layers on top of a channel without obstructing the remaining resin in the interior of the channel.

Even though there are good results when printing closed channels, for the objective of separating particles with the microfluidic device with the spiral, the Asiga resin was not adequate because the relief of the vertical walls of the channel would affect the particle separation. An alternative to reduce the vertical roughness could be to reduce the Z step of the SLA 3D printer.

## 5. Conclusions

In this study, three commercial resins have been tested from the brands Detax, Asiga, and Keyprint, and their width and height resolution were analyzed to produce microfluidic devices.

It was seen that the width resolution is better than the height resolution, and that the printing accuracy is better as the features are bigger and the structures are more separated.

Microfluidic devices were developed using three different approaches: 3D printing a mold to create a PDMS replica, printing the engraved channels, and printing directly the final microfluidic system all in one 3D printing process. We have demonstrated that the absorbance of the resin affects the capability of 3D printing closed channels. As with the resin of higher absorbance, Asiga, it channels with a cover of eight layers of 100 µm per layers and can be printed unobstructed. With the Detax resin, the second on absorbance value, closed channels with covers of up to two layers of 100 µm were reached. On the contrary, for the resin with the lowest absorbance, the Keyprint resin, acceptable results could not be obtained when printing with closed channels. Another conclusion extracted from the absorbance test is that the horizontal roughness is smoother for higher absorbances, while the vertical roughness is smoother for lower absorbances.

To conclude, the design of complex microfluidic circuits using one-step fabrication requires an accurate balance between the printing conditions and the main characteristics of resin, as well as the critical energy required to start the photopolymerization and the optical absorbance coefficient of the resin. A compromise between different parameters such as dimension precision or wall roughness would be necessary for each specific layout.

## Figures and Tables

**Figure 1 polymers-14-02955-f001:**
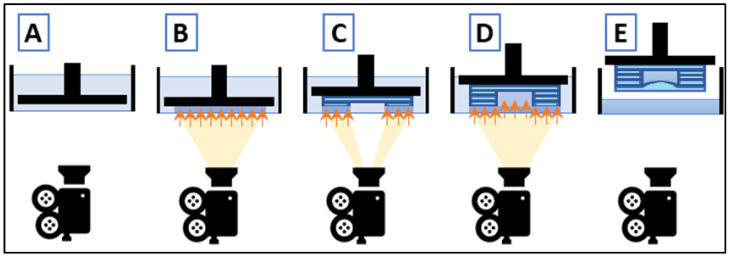
Different phases of printing a cantilever structure. (**A**) The printing platform moves to the end of the vat, where the base of the device will be created. (**B**) The DLP projector exposes all the base. (**C**) Only the channel walls are exposed by the DLP projector. (**D**) Once the channel is created, all the device has to be exposed in order to polymerize a cover for the channel. (**E**) It can be appreciated that the light exposed in (**D**) crossed through the channel and partially polymerized the channel cavity.

**Figure 2 polymers-14-02955-f002:**
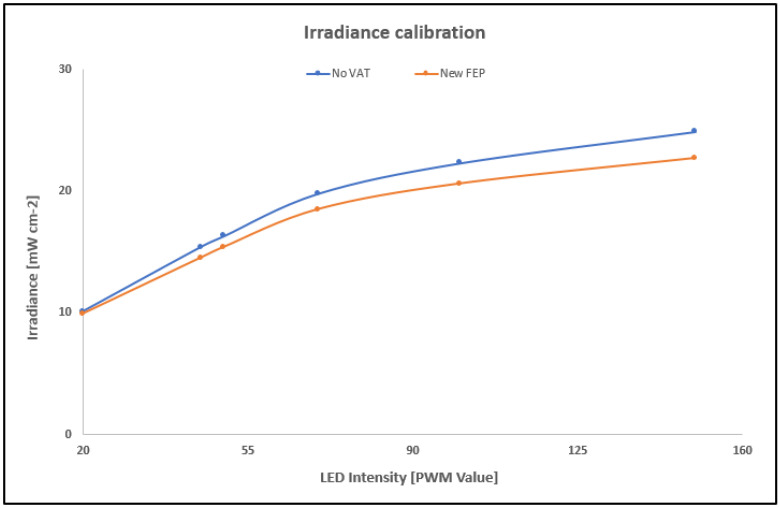
Irradiance calibration results depending on the LED intensity without vat and with a new FEP installed.

**Figure 3 polymers-14-02955-f003:**
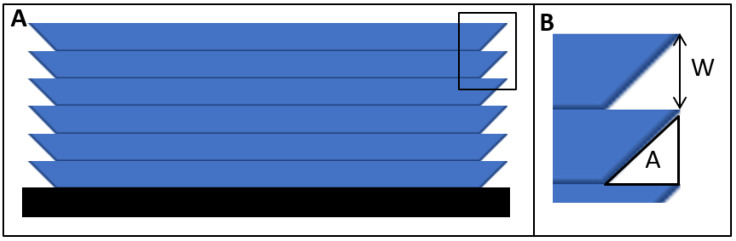
(**A**) Example of a 3D printed part and its sidewall roughness. (**B**) Zoom from 3 (**A**) where the thickness of the layer (W), and the uncured area in every layer (A) are described.

**Figure 4 polymers-14-02955-f004:**
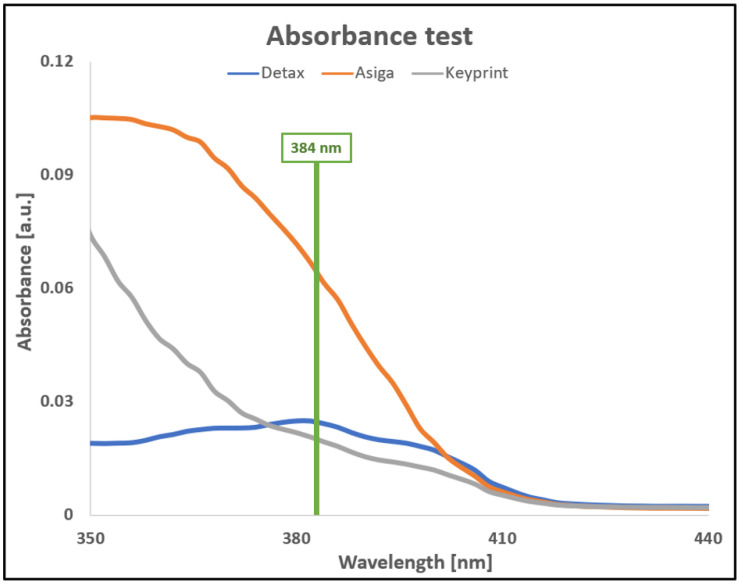
Graph presenting the absorbance for three commercial resins and its dependency with respect to the light wavelength, from 350 nm to 440 nm.

**Figure 5 polymers-14-02955-f005:**
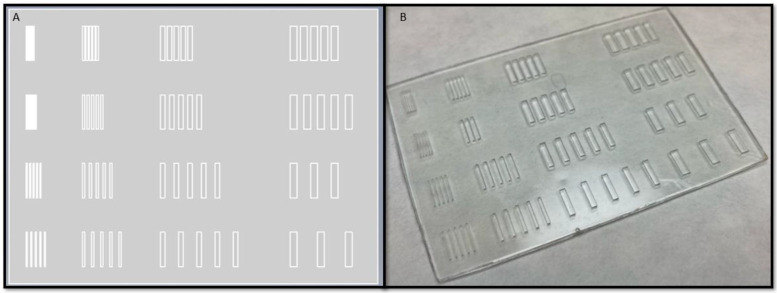
(**A**) Figure of the parametric design made with Inventor from Autodesk. (**B**) Picture of one of the 3D printed piece obtained, for the design of 600 µm height and the resin from Asiga.

**Figure 6 polymers-14-02955-f006:**
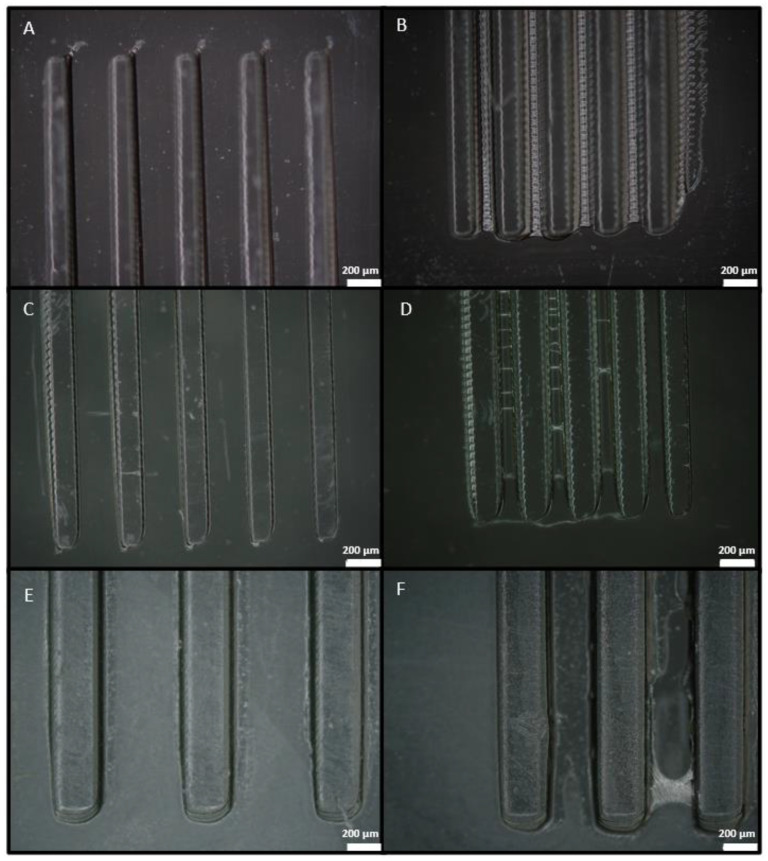
(**A**) Picture of the part printed with Detax, with a designed height of 200 µm, a width of 100 µm with a separation of 300 µm. (**B**) Picture of the part printed with Detax, with a designed height of 200 µm, a width of 100 µm with a separation of 200 µm. (**C**) Picture of the part printed with Asiga, with a designed height of 200 µm, a width of 100 µm with a separation of 300 µm. (**D**) Picture of the part printed with Asiga, with a designed height of 200 µm, a width of 100 µm with a separation of 200 µm. (**E**) Picture of the part printed with Keyprint, with a designed height of 400 µm, a width of 200 µm with a separation of 600 µm. (**F**) Picture of the part printed with Keyprint, with a designed height of 400 µm, a width of 200 µm with a separation of 400 µm. The figures in (**A**,**C**,**E**) are the ones obtained with the minimum height for each resin, the width being half the weight and the separation being three times the width and with a correct definition. The figures in (**B**,**D**,**F**) are the ones obtained with the minimum height for each resin, the width being half the weight and the separation being two times the width, and it can be appreciated that the separation of the structures starts being imperfect. The scale bar for all figures is 200 µm.

**Figure 7 polymers-14-02955-f007:**
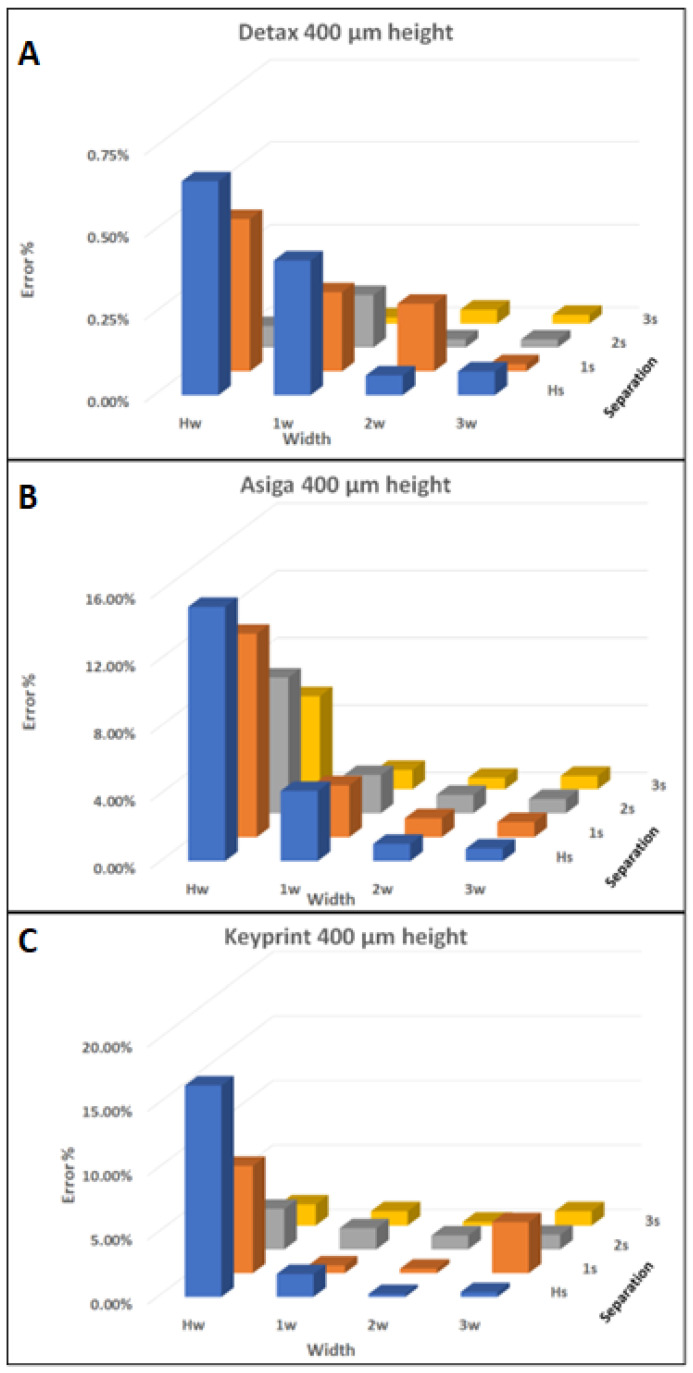
(**A**) Graphic from the error obtained printing the Detax part of 400 µm height. (**B**) Graphic from the error obtained printing the Asiga part of 400 µm height. (**C**) Graphic from the error obtained printing the Keyprint part of 400 µm height. In all (**A**–**C**), the X-axis corresponds to the width of the feature, going from the half of the height in the left to three times the height in the right. The Y-axis instead corresponds to the separation between the features, going from the half of the width in the front to three times the width in the back. The Z-axis represents the error calculated in the height of the features in percentage.

**Figure 8 polymers-14-02955-f008:**
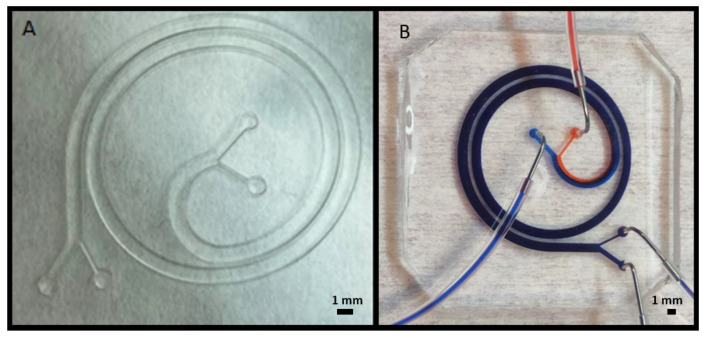
(**A**) 3D printed result with Detax resin of the extruded spiral, with a designed channel width of 1 mm and height of 600 µm. (**B**) Picture of the microfluidic system created with the PDMS replica based on the 3D printed part of (**A**).

**Figure 9 polymers-14-02955-f009:**
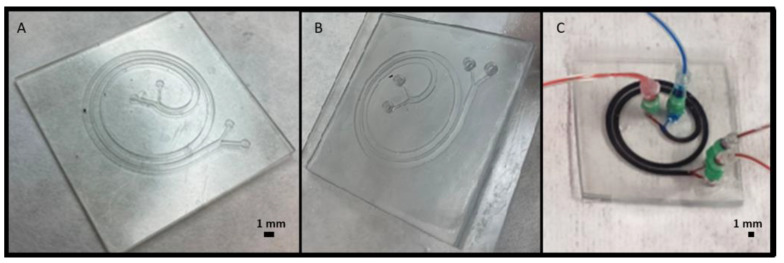
(**A**) 3D printed result with Detax resin of the embossed spiral, with a designed channel width of 1 mm and height of 600 µm. (**B**) Assembly of an open spiral and a cover printed separately, with Keyprint resin. (**C**) Microfluidic system consisting of two inputs and two outputs, used to separate particles by their size.

**Figure 10 polymers-14-02955-f010:**
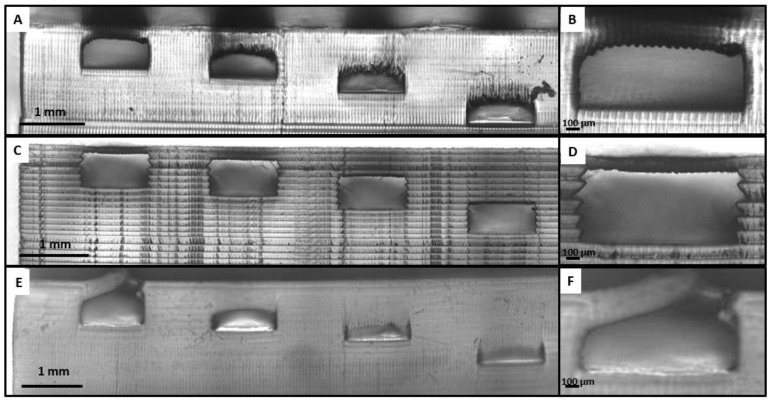
Pictures of the 3D printed closed channels using Detax (**A**), Asiga (**C**), and Keyprint (**E**) resins. All the structures in (**A**,**C**,**E**) consist of four channels of 1 mm width and 500 µm height with a different number of layers in the cover of the channel (1, 2, 4, 8 layers of 100 µm each from left to right). A zoomed view of the channel with a cover of 100 µm printed can be also seen with the Detax (**B**), the Asiga (**D**), and the Keyprint (**F**) resins.

**Figure 11 polymers-14-02955-f011:**
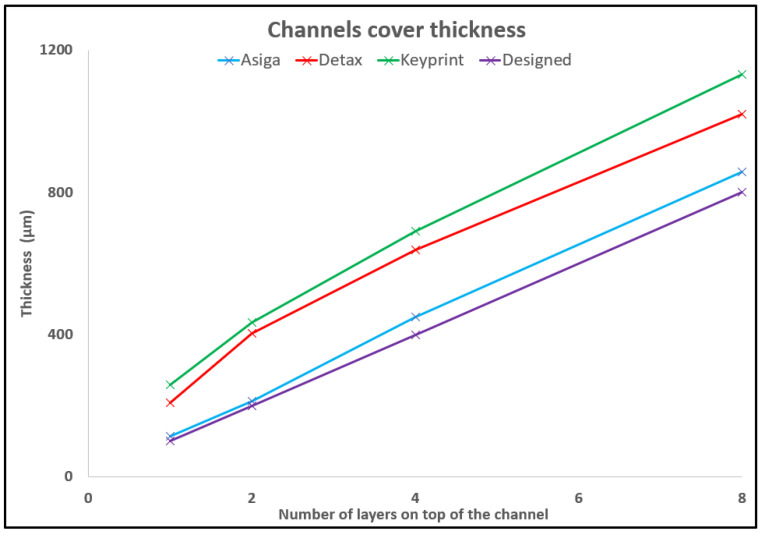
This graphic shows the thickness of the cover of every channel printed with the Asiga, Detax, and Keyprint resins vs. the design file values for one, two, four, and eight layers, respectively. The designed cover had a nominal thickness of 100 µm, 200 µm, 400 µm, and 800 µm.

**Figure 12 polymers-14-02955-f012:**
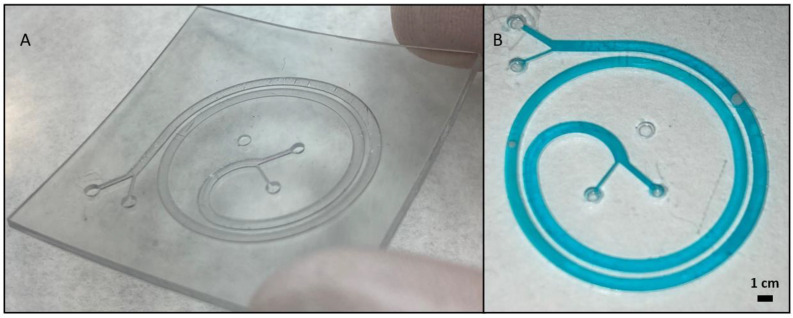
(**A**) Microfluidic chip printed in one single process. 3D printed result with Detax resin of the spiral, with a designed channel width of 1 mm and height of 600 µm. Only one layer of 100 µm has been printed on top of the channel to not obstruct the channel. (**B**) Test that shows there is no leakage with water tinted with blue colorant.

**Table 1 polymers-14-02955-t001:** Optimal settings empirically determined for the different resins tested.

Variable	Detax	Asiga	Keyprint
Exposure time for burn in layer (s)	6	40	7
Exposure time for normal layer (s)	2	20	4
Layer thickness (µm)	100	100	100
Light power (PWM)	35	35	35

**Table 2 polymers-14-02955-t002:** Height error for every resin type and every different height design.

Resin	Designed Height (µm)	Average Height Error
Detax	600	7.14%
	400	9.35%
	200	22.32%
Asiga	600	16.73%
	400	17.39%
	200	27.25%
Keyprint	600	34.43%
	400	46.38%

**Table 3 polymers-14-02955-t003:** Width and height error for extruded and embossed spiral printed with the three different types of resin.

Resin	Feature Type	Width Error	Height Error
Detax	Extrusion	0.14%	0.94%
	Embossed	0.30%	3.62%
Asiga	Extrusion	0.24%	0.93%
	Embossed	0.31%	4.98%
Keyprint	Extrusion	0.34%	3.40%
	Embossed	1.43%	4.67%

## Data Availability

Data is contained within the article.

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
