# Peer review of "Development of a Custom-Made 3D Printing Protocol with Commercial Resins for Manufacturing Microfluidic Devices"

_polymers, 2022, doi:10.3390/polym14142955_

Round 1
Reviewer 1 Report
The article submitted by Subirada et al. (polymers-1793117), entitled “Optimization of a custom-made 3D printing protocol with commercial resins. Application to the manufacturing of microfluidic devices.”, investigated the digital light processing (DLP) 3D-printed microfluidics. There methods, which fabricating the microfluidic device directly or indirectly, were achieved and compared. The topic is interesting, and it can be classified as a typical application of polymer additive manufacturing. However, mechanism analysis is better to be strengthened in this article for improving its impact. For the benefit of the reader, this reviewer would like to suggest a revision before its publication.
1) The shapes and sizes of the samples were evaluated. However, the mechanical properties of the 3D printed microfluidics are better to be evaluated, if possibly, since the low mechanical properties of SLA samples are common.
2) The different exposure times of the three resins were found, as shown in Table 1. The reason of the different solidification behavior for the three resins is better to be disclosed.
3) The three methods of obtaining the microfluidics should be furtherly compared, especially the structural difference of the fabricated components.
4) To improving the impact of this article, a further analysis of orientation for additive manufactured samples was suggested to discuss. Maybe, for DLP process, it is not obviously. However, it is still worthy of paying attention. About this point, an article was recommended, Polymer Composites, 2020, 41, 60-72, although the used materials and methods were different. Authors may mention it and give a simple introduction about this concern.
5) In figure 1, the types of the line are better to be revised for making a distinction.
6) Improving the quality of figure 6 is somewhat necessary.
In conclusion, some revisions are suggested before its publication.
Reviewer 2 Report
This article describes about the 3D printing for manufacturing the micro fluid devices. In general, the experiment quantity is enough for a research paper. However, the discussion is still very rough. So, I suggest authors modify the paper with these issues:
1. The title mentions about the “Optimization”, so, the content has to show out the optimization, or any idea about the “Optimization”
2. All results must have the discussion (why? effect of resin, application….)
3. For all figure names: The description of Figure name should be in the paper content
4. Part 2.2 should explain the reason for selecting these three resins (Detax, Keyprint, and Asiga) for researching in this article (is there any special between these resins?)
5. What is the role of Figure 2?
6. Part 3.1 mentions about the result, so, it should have the conclusion or any discussion about these result
7. Part 3.2, line 270 – 271 should determine the “three different resin”
8. Part 3.2, line 279-280: paper should mention about the reason for selecting three parameters for researching
9. Part 3.3 must have the discussion
10. Line 402: what is the Keystone resin? Authors should define it
11. Line 102 – 403: what does this data use for ?
12. Line 423 – 424: discuss / explain about this result. What is the application of this result ?
13. Figure 11:data of “Designed” did not describe in the content
14. Line 477-478: Which equipment was used for measuring this result ?
15. Line 507 – 508: “…only one layer was being printed on top of the channel,…” What is the limitation of this method in manufacturing the micro fluid device?
16. Part 4 should be mixed with other parts
17. The micro fluid module in Part 3.4 should have a Figure mention about the dimension / size
18. In general, the paper structure should be mkodified for increasing the link between parts
Round 2
Reviewer 1 Report
The revised version is better than before. I recommend its publication in present form.
Author Response
Thank you for the first revision

Reviewer 2 Report
In general, the paper quality has a clearly improvement, also, the researching data was described clearer than the old version. This version is good for publishing after these modifications:
-Paper title: should be one sentence
- What is the role of Equation 3 (line 278 – 281) in this paper?
- The area if Figure name should not describe or discuss anything. These discussions should be in the paper text. Authors check all Figure name (as: Fig. 7, Fig 10…)
- Line 554 – 556: Check the paragraph again, this paragraph has only one sentence
- English checking: especially with the modification part of this version
